# Efficient agricultural drip irrigation inspired by fig leaf morphology

Shijie Liu [1,2,3], Chengqi Zhang[1,3], Tao Shen[1], Zidong Zhan[1,3], Jia Peng[1,3], Cunlong Yu[1,3], Lei Jiang [1,2,3] & Zhichao Dong [1,3] ✉

Irrigation is limited by water scarcity. Here, we show how a drip irrigation system inspired by the leaf of the fig tree *Ficus religiosa* (also known as the bodhi tree) can improve irrigation efficiency. The reverse curvature of the leaf regulates the convergence process of multiple water streams, while its long-tail apex allows for fast water drainage with the droplet separation centroid beyond the leaf apex. We explain why drip frequency increases after the break-up of contact line pinning at the apex tip by using scaling laws for drip volume and analyzing drainage dynamics. We build a drip irrigation emitter inspired by the bodhi leaf apex and compare the germination efficiency of wheat, cotton, and maize under different irrigation modes. These results show that the proposed bodhi-leaf-apex-mimetic (BLAM) drip irrigation can improve water saving while ensuring germination and seedling growth.

Irrigated agriculture accounts for 70% of all freshwater withdrawals globally[1]. Water scarcity severely affects nearly 4.0 billion people and is projected to rise, where the expansion of irrigated agriculture is listed as one of the main driving issues[2]. Ongoing drought poses a significant threat to agricultural water supply[3–5]. Farmers in Central Asia almost drained a giant inland lake primarily due to agricultural water overuse[6]. Moreover, Europe is experiencing its worst drought in at least 500 years, where drought limits the water supply for agricultural irrigation[7]. In the context of climate change, drought is one of the most limiting factors influencing crop production. Maize, for example, is highly vulnerable to water deficit, which causes a significant yield loss of nearly 40%[8]. Drought puts livelihoods at risk, often halting and reversing gains in food security and poverty reduction and hampering efforts to reach Sustainable Development Goals 1 and 2[9]. Therefore, developing precise agricultural irrigation methods, such as drip irrigation, shall be beneficial to resolving the water shortage issue[10].

Besides the drought area, controlled water delivery is of great necessity for the survival of plants in tropical rainforest regions[11,12], where the competition of vast numbers of plant species and limited natural resources leads to fierce competition for acquiring or shedding water[12–16]. Many plant leaves utilize various structural features to manipulate the dropwise transport and separation behaviors, and intriguing examples include the directional spread of water condensation on the peristome of Nepenthes[13,14], enhanced water drainage on the leaf apex of Alocasia macrorrhiza[15], and liquid direct steering on the Araucaria leaf[16]. Analyzing natural designs is therefore beneficial for the answers to the problems of precise irrigation[17,18]. Among these structures, the long-tail leaf apex of *Ficus religiosa* (bodhi tree), a classical characteristic of rainforest and famous for the Buddhism association, benefits leaf drying[19], which decreases the need for leaf support and reduces organism colonization[20]. How such a long-tail leaf apex regulates fast water-shedding of bodhi leaf is attractive yet elusive.

Here, we demonstrate the bodhi leaf apex's reverse curvature and long-tail enhanced drainage mechanism. Water drainage with high dropwise dripping frequency and low dispensed volume can be realized by simply designing a leaf apex structure with a reverse curvature of 0.618, the golden section ratio. The long tail of the bodhi leaf can realize high drip frequency by breaking the contact line pinning at the apex with the centroid of the newly forming drop beyond the apex tip, changing the drip model from the Above-drip state to the Beyond-drip state. Optimal bodhi leaf apex is applied to the agricultural drip irrigation apparatus. Compared with general methods, including border irrigation, wasting water, and spray or drip irrigation that squeezes

[1]CAS Key Laboratory of Bio-inspired Materials and Interfacial Sciences, Technical Institute of Physics and Chemistry, Chinese Academy of Sciences, 100190 Beijing, China. [2]Suzhou Institute for Advanced Research, University of Science and Technology of China, 215123 Suzhou, Jiangsu, China. [3]School of Future Technology, University of Chinese Academy of Sciences, 100049 Beijing, China. ✉e-mail: dongzhichao@mail.ipc.ac.cn

droplets from small nozzles, suffering high flow resistance and nozzle blockage, our state-of-the-art biomimetic dripper demonstrates that the reverse curvature and the long tail ensure precise and controllable drop emission. All advantages benefit water saving, fast sprouting, and upright growth of crop seedlings, promising future precise agriculture.

## Results

### Reverse curvature and long-tail enhanced water drainage efficiency

The *Ficus* genus is a typical rainforest species with drip tips as its survival strategy. It mainly lives in the tropics and subtropics, remarkably diverse in southeast Asia with high humidity (Fig. 1a)[21,22]. Eighteen species (Supplementary Fig. 1) in the *Ficus* genus were investigated and classified into five groups based on apex shapes[23], i.e., round (*Ficus curtipes*), obtuse (*Ficus altissima*), acute (*Ficus concinna*), acuminate (*Ficus elastica*), and caudate (*Ficus religiosa*), as sketched in the bottom of Fig. 1a. Effective water drainage entails the surface morphology or structure to break the contact line pinning at the leaf apex to form dropwise dripping. Drainage efficiency[15] is found to be determined by the drip frequency $f$ (Hz) and the drip volume $V_d$ (μL). The $f$ and $V_d$ were compared on these leaves' apices under the same inclination angle $\beta$ of 60° and volumetric flow rate $Q$ of 8.0 mL min$^{-1}$ (Fig. 1b). The long-tail caudate apex of the bodhi leaf exhibits the best drainage efficiency with $f$ and $V_d$, -2.5 and -0.4 than those of round apex of *Ficus curtipes*, respectively (Fig. 1c, d).

The natural bodhi leaf is overall flat with a curved leaf margin (Supplementary Fig. 2). The curved bodhi leaf margin is found to have a three-centered arch, like the classical "Ogee" arch[24], with a reverse curvature (Fig. 1e). On a rainy day, rainwater droplets impact the leaf surface, and several rainwater streams converge and merge into a single water flow, which drains quickly along the leaf apex (Supplementary Fig. 3 and Supplementary Movie 1). The convex curvature of the leaf body part is fitted by a circle and defined as $1/R$, the concave curvature of the leaf base part $1/r$, and the reverse curvature $r/R$ (Fig. 1e and Supplementary Fig. 4). Measuring 76 bodhi leaves, we found that the $r/R$ approximately exhibits a normal distribution between 0.1 and 1.0 with a mean value of 0.6 (Fig. 1f). Moreover, the space in between the two circles at the base of the leaf tip is defined as the base width $W_{base}$. The length of the long-tail $L_{apex}$, the distance between the $W_{base}$ and the tip of the leaf apex, occupies about 18–40% of the total leaf length $L_{total}$ (Fig. 1g). The $L_{apex}/W_{base}$ is between 4.6 and 13.0 and increases linearly with the $L_{apex}/L_{total}$. Although the long-tail caudate is famous for its drip-tip, the symbol of rainforest, and the key feature of the Sri Lanka flag, the fast drainage mechanism remains a mystery. We hypothesize that the surface curvature shape[25], including both leaf margin curvature and long-tail tip, matters in water drainage efficiency. That is, the reverse curvature ($r/R$) regulates the convergence process of multiple water streams, and the long tail ($L_{apex}/W_{base}$) dominates the drainage dynamics of water flow on the leaf apex.

### Curvature reshaped flow hydrodynamics and drop separation centroid

To reveal the drainage mechanism, we constructed artificial polyethylene terephthalate (PET) leaves that mimic the natural bodhi leaf by laser cutting (Fig. 2a, see "Methods"). The margin of the PET-based artificial bodhi leaf was bulged, forming a ridge with a width of -80 μm (Fig. 2a, inset) due to the fusing effect in the laser-cutting process. Such a ridge functions to resist water flow without overflow[26] and pin the three-phase contact line at the PET-based leaf apex tip. Based on the leaf shape analysis in Fig. 1e–g, we divide the bodhi leaf model into two functional regions, that is, the convergence region whose shape is dominated by the reverse curvature $r/R$ (Fig. 2a–e and Supplementary Fig. 5) and the drainage region dominated by the long tail $L_{apex}/W_{base}$ (Fig. 2f, g). The experimental setup is shown in Fig. 2b, f and consists of PET-based artificial bodhi

leaves with $r/R$ ranging from 2/21 to 21/21 (Fig. 2d) and $L_{apex}/W_{base}$ ranging from 2.5 to 15.0 (Fig. 2g).

First, a three-needle experimental setup (Fig. 2b) was designed to explore the effect of the reverse curvature, $r/R$, on the convergence of multiple water flows in the convergence region (Fig. 2b–d). The $r/R$ mean value of collected natural bodhi leaves is -0.6 (Fig. 1f), close to the golden section point 0.618[27], which is directly tied to a numerical series known as the Fibonacci sequence, "*1, 1, 2, 3, 5, 8, 13, 21, …*". The curve shape with $r/R = 0.618$ is known as a golden spiral[27]. It can be found throughout nature, most prominently in seashells, *Romanesco* cauliflowers, pinecones, and sunflower seed heads[28–31]. The number of 0.618 is approximately equal to the ratio of any preceding number of the subsequent one (for example, $13/21 \approx 0.618$) in the Fibonacci sequence. We therefore designed a set of $r/R$ values as 2/21, 5/21, 8/21, 13/21, and 21/21, to examine the effect of the reverse curvature $r/R$ on drainage dynamics of bodhi leaf apex (Fig. 2d, e). Water was injected off a three-orifice needle (inner diameter, 0.5 mm) located just above the convergence region on the PET-based leaf plate at the injection flow rate $Q$ in the range of 8.0–36.0 mL min$^{-1}$. Cameras recorded water flow and drainage phenomena from both top and side views (Fig. 2c and Supplementary Fig. 6). Motion analysis software measured the drop separation site, the drop centroid, the drip frequency $f$, and the water flow speed.

Figure 2c shows the drainage process of water flow at an inclination angle $\beta$ of 30° and the flow rate $Q$ of 32.0 mL min$^{-1}$. We first focused on the water drainage dynamics at $r/R$ of 0.618. Three water streams converged along the leaf margin into one stream and flowed toward the leaf tip in a stable water film on the leaf apex. Such convergence behavior aggregates water streams and makes the converged water flow breaking contact line pinning[32] at the apex tip. The Rayleigh-Plateau (R-P) instability was suppressed, and a nearly smooth water flow on the leaf apex was obtained (Fig. 2c, top). The centroid of the newly forming drop was beyond the leaf apex. This drainage process was termed the Beyond-drip state (Supplementary Movie 1). The droplet experienced a sudden instability induced by the increased gravitational force component. It decreased resistant capillarity beyond the leaf apex (from steps I to II in Fig. 2c, top). The reconstructed balance forced the water droplet to drip off with high frequency and small volume. As shown in Fig. 2e, the $f$ jumps from 22.7 to 40.3 Hz when we increase the $r/R$ from 2/21 to 13/21 (Fig. 2c, e, Supplementary Fig. 5).

At a smaller $r/R$ value ($r/R = 0.238$), the two rivulets from the left and right sides collided in the convergence region (Fig. 2c, bottom). The contact line pinning along the leaf margin resisted the water flow without overflow, and the surface curvature reshaped the flow behavior, where water flow accumulated. It rose in height at the center of the convergence region (Supplementary Movie 1). The kinetic energy loss during the water rivulets collision in the convergence region led to decreased water flow speed in the drainage region. Particle image velocimetry[33] evaluated the water flow speed $v$ on the leaf apex with $r/R = 0.618$ and $r/R = 0.238$ (Supplementary Fig. 6, Supplementary Movie 1, see "Methods"). Setting the same initial water flow condition, we found that the measured $v$ on the leaf apex with $r/R = 0.618$ was $288 \pm 18$ mm s$^{-1}$ from the side view, which was much faster than that, $217 \pm 22$ mm s$^{-1}$ on the leaf apex with $r/R = 0.238$ (Supplementary Fig. 6, Supplementary Movie 1). Moreover, the contact line pinning at the leaf apex could slow the flow without flowing out of the leaf apex (Fig. 2c, bottom). Such a sudden accumulation of water triggered the R-P instability of the water flow on the apex (the drainage region)[34] and made the water flow break up into bigger droplets[35] with the centroid of the newly forming drop above the apex tip (Fig. 2c, bottom). We termed this drainage behavior as the Above-drip state. As a result, the dropwise separation occurred on the leaf apex with a low drainage frequency $f$ that reduced to 26.0 Hz, only -2/3 of which in the $r/R = 0.618$ case (40.3 Hz).

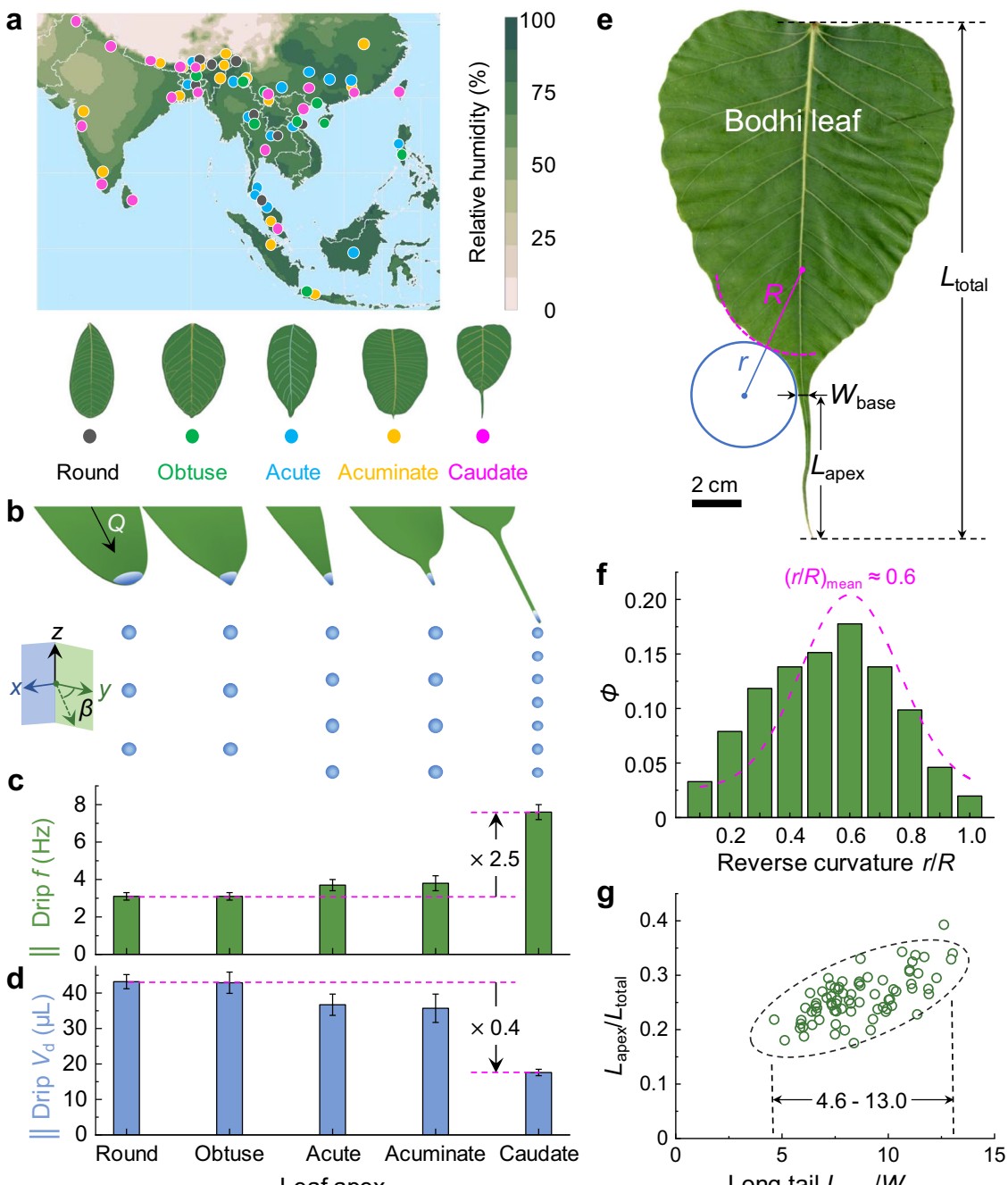

**Fig. 1 | Reverse curvature and long tail of bodhi leaf enhanced water drainage efficiency. a** Geographical distribution of five chosen species of the *Ficus* genus and average annual relative humidity in southeast Asia. The relative humidity map is reproduced[21]. Geographical distribution data of the *Ficus* genus is extracted from GBIF[22]. Bottom schematics are round, obtuse, acute, acuminate, and caudate leaf apex shapes. **b** Sketch of water drainage on demonstrated apex shapes at the same injection flow rate $Q$ of 8.0 mL min$^{-1}$ and the incline angle $\beta$ of 60°. Drip frequency $f$ (**c**) and drip volume $V_d$ (**d**) of water droplets dripping from the apices. The caudate apex of bodhi leaf has the highest $f$ and smallest $V_d$, revealing the highest drainage efficiency. **e** Optical image of a bodhi leaf with shape parameters, the radius of curvature along leaf margin $R$ and $r$, the base width of leaf apex $W_{base}$, the length of

apex $L_{apex}$, and the total leaf length $L_{total}$. The statistics of reverse curvature $r/R$ (**f**), long tail $L_{apex}/W_{base}$, and $L_{apex}/L_{total}$ (**g**) are based on the measurements of 76 natural bodhi leaves. In (**f**), $\Phi$ means the probability density of $r/R$ in the statistics, and the purple dashed curve was the Gaussian fitting of the $r/R$ histogram. Note that the left and right $r/R$ values of natural bodhi leaves were separately measured and collected together. Thus, there were 76 × 2 = 152 leaves counted up in (**f**). The mean value of $r/R$ is -0.6. In (**g**), the black dashed ellipse was 95% confidence ellipse for the linear fitting. Data in (**c**) and (**d**) are shown as mean ± SD, and the error bar represents SD ($n$ = 3 independent experiments). Source data for (**c**, **d**, **f**, **g**) are provided as a Source Data file.

Increasing the $r/R$ above 0.618, we found that the drainage dynamics were in the Beyond-drip state (Fig. 2e) and the water flow tended to break away from the leaf margin (see red arrows in Supplementary Fig. 5). The variation of drainage frequency $f$ revealed in Fig. 2e pointed to a critical reverse curvature $r/R$ of

0.618, a natural tailor[29,30], and consistent with the statistical mean value of real bodhi leaves shown in Fig. 1f. The $r/R$ of 0.618 is, therefore, the optimal shape parameter for effectively converging water flow to enhance drainage efficiency with less kinetic energy loss.

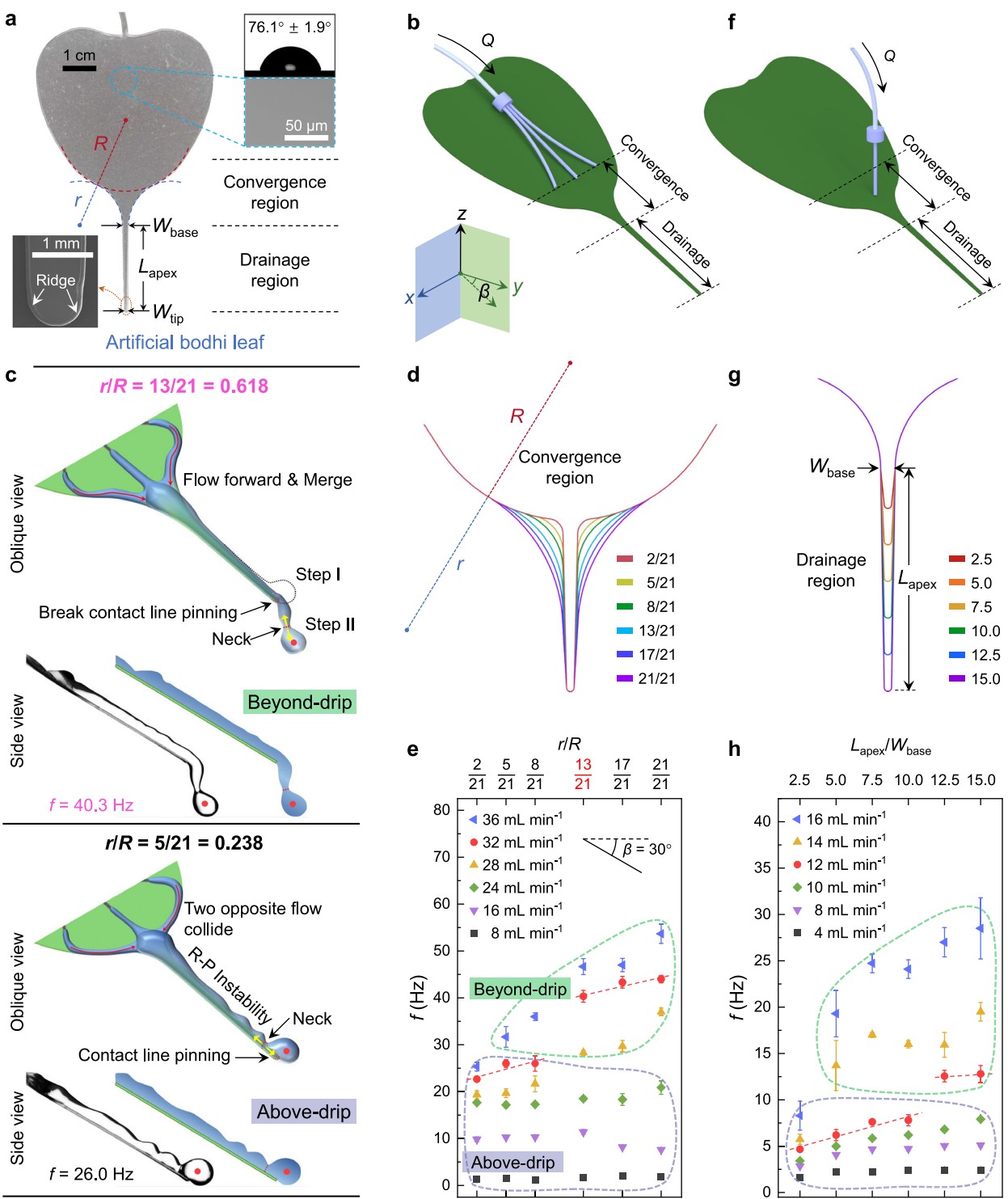

**Fig. 2 | Curvature and long tail of artificial bodhi-leaf reshaped drop separation centroid and enhanced water drip frequency. a** Optical image of PET-based biomimetic bodhi leaf with shape parameters $R$, $r$, $W_{base}$, $L_{apex}$, and $W_{tip}$ noted from top to bottom. Reverse curvature, $r/R$, controls convergence region and long tail, $L_{apex}/W_{base}$, controls drainage region. Insets are the contact angle of a water droplet on the PET surface and scanning electron microscopy (SEM) images of the PET substrate in the central part and at apex tip. **b** Scheme of the three-needle experimental setup. **c**, **d** Effect of the reverse curvature $r/R$ on the aggregation of water flow. **c** Oblique and side views of water flow dynamics on artificial bodhi leaf with $r/R = 0.618$ (the golden section point) where water drop separates beyond the artificial leaf apex, and $r/R = 0.238$ where water drop separates above the leaf apex, at the same injection flow rate $Q$ of 32.0 mL min⁻¹. Red arrows in the oblique views indicate water flow direction, and red dots in the side views indicate the centroid of the newly forming drop near the apex tip. **d** $r/R$ accords with the Fibonacci sequence and ranges from 2/21 to 21/21. **e** Water drainage frequency $f$ of various $r/R$ values at 8.0–36.0 mL min⁻¹ and $\beta = 30°$. The light-blue and mint-green dashed curves denote Above-drip and Beyond-drip states. **f** Scheme of the single-needle experimental setup. **g** Long-tail $L_{apex}/W_{base}$ ranges from 2.5 to 15.0. **h** Water drainage frequency $f$ of various $L_{apex}/W_{base}$ at 4.0–16.0 mL min⁻¹ and $\beta = 30°$. Note that some error bars are smaller than the symbols. The red dashed lines in (**e**) and (**h**) were linear fitting of corresponding data points. Data in (**e**) and (**h**) are shown as mean ± SD, and the error bar represents SD ($n = 3$ independent experiments). Source data for (**e**, **h**) are provided as a Source Data file.

## Long tail enhanced water drip frequency

Next, a systemic investigation was performed to reveal the effect of long-tail (drainage region) on drainage efficiency (Fig. 2f–h). A single-needle experimental setup was designed to avoid the influence from the convergence region, where the ejected flow was localized just above the drainage region (Fig. 2f). The long tail $L_{apex}/W_{base}$ ranges from 2.5 to 15.0 (Fig. 2g, h). It covers the statistics values of real bodhi leaves shown in Fig. 1g. Considering the energy supply toward the leaf apices is difficult for natural plants, we hypothesize that the optimized parameter $L_{apex}/W_{base}$ would be beneficial for the enhanced drainage as well as the reduction of damage or wilting of long tail[12]. Taking $Q = 12.0$ mL min$^{-1}$ as an example, $f$ increased with the $L_{apex}/W_{base}$ and stabilized when $L_{apex}/W_{base}$ reached 12.5 (Fig. 2h, Supplementary Movie 2). When $L_{apex}/W_{base} \leq 10.0$, water drainage dynamics on apices were in the Above-drip state with $f \leq 8.0$ Hz. In contrast, when $L_{apex}/W_{base} \geq 12.5$, water drainage dynamics on apices developed into the Beyond-drip state with $f \geq 12.0$ Hz (Supplementary Fig. 7), increasing at least 1.5 times in case of 12.0 mL min$^{-1}$.

Taking together the effect of convergence and drainage regions, we investigated the relationship between the drainage frequency $f$ and $L_{apex}/W_{base}$ by using both the three-needle and single-needle setups (Supplementary Fig. 8). We found that $f$ both increased with $L_{apex}/W_{base}$ and reached a plateau value after $L_{apex}/W_{base}$ reached 12.5 (Supplementary Fig. 8b). Moreover, the threshold flow speeds of water flows that break the contact line pinning at the apex are the same for both three-needle and single-needle experiments (Supplementary Fig. 9, see "Methods"). Therefore, the optimized bodhi leaf apex parameters include $r/R$ of 0.618, $L_{apex}/W_{base}$ of 12.5, and $W_{tip}$ of 1.0 mm. Besides shape parameters, the effects of leaf apex wettability on drainage behaviors were also explored (Supplementary Fig. 10). The suitable contact angle (CA) range for continuous and controllable drainage was found to be 30–110°. In the following experiments, we will adopt PET-based leaf apices with intrinsic wettability and optimized parameters and use the single-needle experimental setup to guarantee a stable and controllable drainage process for further discussion.

## Beyond-drip state stabilizes deviation with reduced drip volume

We moved on to exploring the effects of flow rate, $2.0$ mL min$^{-1} \leq Q \leq 20.0$ mL min$^{-1}$, on the drainage process at $\beta$ of 30–60° (Fig. 3a and Supplementary Fig. 11a). Water drainage behaviors changed from the Above-drip to the Beyond-drip and then to the Beyond-jet, as $Q$ increased from 2.0 to 20.0 mL min$^{-1}$ (Fig. 3a and Supplementary Fig. 11b), and drip volume $V_d$ decreased gradually. Taking $\beta = 30°$ as an example, $V_d$ in the Beyond-jet state, ~2 µL, was only 0.07 times than that of the Above-drip state (~28 µL).

In the Above-drip state, the centroid of the newly forming water droplet was above the apex tip (see $\beta = 30°$, $Q = 8.0$ mL min$^{-1}$ in Supplementary Fig. 11b). The drip frequency $f$ was in good linear relation with $Q$ (Supplementary Fig. 11c, d), and the drip volume $V_d$ kept constant during the flow rate of 2.0–8.0 mL min$^{-1}$ at $\beta$ of 30° (Supplementary Fig. 11e). As shown in Fig. 3b, the contact line pinning at the apex tip stably resisted the dripping of accumulated droplets until reaching a critical drop volume, $V_d$. Once the drop was emitted, the neck retracted entirely back onto the apex tip, and a new ridge grew at the apex tip and formed a droplet of the same size as the previous one, thus making the drop emission process periodic and monodisperse (Supplementary Movie 3). Balancing drop gravity and the contact line pinning, $\rho \cdot V_d \cdot g \cdot \sin\beta = k \cdot \gamma \cdot (l_{arc} - W_{tip} \cdot \cos\theta_a)$[36], results in the drip volume $V_d$ of $\frac{k \cdot \gamma \cdot l_{arc}}{\rho \cdot g \cdot \sin\beta} - \frac{k \cdot \gamma \cdot \cos\theta_a}{\rho \cdot g \cdot \sin\beta} \cdot W_{tip}$. Here $\gamma$ is the water surface tension, $\rho$ is the water density, $g$ is the gravity acceleration, $k$ is the coefficient factor that relates to the shape of the contact line[37] and is calculated to be 0.93 (Supplementary Fig. 11e), $\theta_a$ is the advancing contact angle (CA), and $l_{arc}$ is the arc length of neck, respectively. The calculated $V_d$ values (Supplementary Fig. 11e) corresponded with the

experimental $V_d$ results ($V_d = Q \cdot f^{-1}$, when the relative deviation of dripping time interval was less than 5%, see "Methods"). The effect of apex tip width $W_{tip}$ on drip volume $V_d$ under $\beta$ of 30° was then explored to validate this formula. Cu-based leaf apices fabricated by laser-cutting were chosen to ensure the accuracy of the apex tip parameters (Supplementary Fig. 12). The stable Above-drip state entails $W_{tip} \geq 1.0$ mm, and the experimental results fit the equation line well (Fig. 3c, d). $W_{tip}$ of 1.0 mm is thus regarded as the minimum and optimal tip width.

Increasing $Q$ to 12.0–15.0 mL min$^{-1}$ under $\beta$ of 30°, we found that the water flow broke the contact line pinning at the apex tip, and the centroid of the newly forming drop was beyond the apex tip, i.e., the Beyond-drip state. The inertial force within the water flow prevented the liquid neck from retracting entirely onto the apex tip after the droplet pinched off[38]. It forced water droplets to form beyond the leaf apex (Supplementary Movie 3). The reconstructed balance of the increased flow inertia, the increased gravitational force component, and the decreased resistant capillarity at the separation neck governed the Beyond-drip droplet to drip off with a smaller volume $V_d$ of $-v/g \cdot Q + A \cdot \gamma/\rho g \cdot W_{tip}$ than for the Above-drip droplet (Fig. 3e). In this formula, $v$ is the water flow velocity along the apex and $A$ is the coefficient factor which relates to the water column shape at the apex tip. The coefficient factor $A$ was found to be a constant of ≈2.9 according to the linear-fitting results of three $\beta$ values (Fig. 3f). Taking $\beta = 30°$ as an example, the water flow velocity $v$ based on the linear-fitting slope was calculated to be 0.25 m/s, close to the measured $v$ at $Q$ of 12.0 mL min$^{-1}$ under $\beta = 30°$ (see Supplementary Fig. 9f). Overall, the slope of $V_d$ - $Q$ in the Beyond-drip state was controlled by the flow inertia and the gravity (Fig. 3f)[35], which also explained why the drip volume $V_d$ started to decrease at high $Q$ in the Above-drip state (Supplementary Fig. 11e).

At high flow rates ($Q \geq 16.0$ mL min$^{-1}$, $\beta = 30°$), the water flowed out of the leaf apex tip, forming a column of liquid jet with length $L_d$, i.e., the Beyond-jet state (Fig. 3g and Supplementary Movie 3). Triggered by R-P instability, the fluid jet broke into tiny droplets beyond the apex. The relationship between the pinched droplet diameter $D$ and initial filament diameter $D_0$ can be written as $D = (3\pi/2\eta)^{1/3} \cdot D_0$, where $\eta$ is the relative viscosity of water and air[39]. The experimental results fit well with the theoretical equation (Fig. 3h). The pinched droplet diameter $D$ decreased slightly with $\beta$. Moreover, $L_d/D$ was considered a key criterion for the definition of transition from dripping to jetting[40], and here for the description of the transition from Beyond-drip region ($L_d/D < 2$) to Beyond-jet region ($L_d/D \geq 2$) (Supplementary Fig. 13).

Figure 3i compared the relative deviation of drip volume $V_d$, $(V_d - \overline{V_d})/\overline{V_d}$, of continuous ten droplets in one drainage process at a series of $Q$ values ($\beta = 30°$). The relative deviation was less than 10% in the Above-drip and Beyond-drip states. Moreover, the Beyond-drip state had a smaller $V_d$ than that of the Above-drip state (Fig. 3a). $V_d$ was about 2.0–5.0 µL in the Beyond-jet state, smaller than the other two drainage states, but with the most significant relative deviation (Fig. 3i). In short summary, balancing the drip volume, deviation, and drip frequency, the Beyond-drip state can achieve a smaller $V_d$, minor relative deviation and higher frequency, simultaneously, promising the need for the precise small droplet production. Considering it is a facile way to control the dripping model by designing leaf apex, we will demonstrate the proof-of-concept applications in the following discussion.

## Natural bodhi-leaf-apex-mimetic (BLAM) design for drip irrigation

Agricultural irrigation consumed ~70% of global freshwater resources during the past century[41]. Although irrigation resolves spatial and temporal disconnects between water supply and water demand and allows us to grow crops in semideserts, eternal vigilance, in many places worldwide, water is still wasted through inefficient, traditional

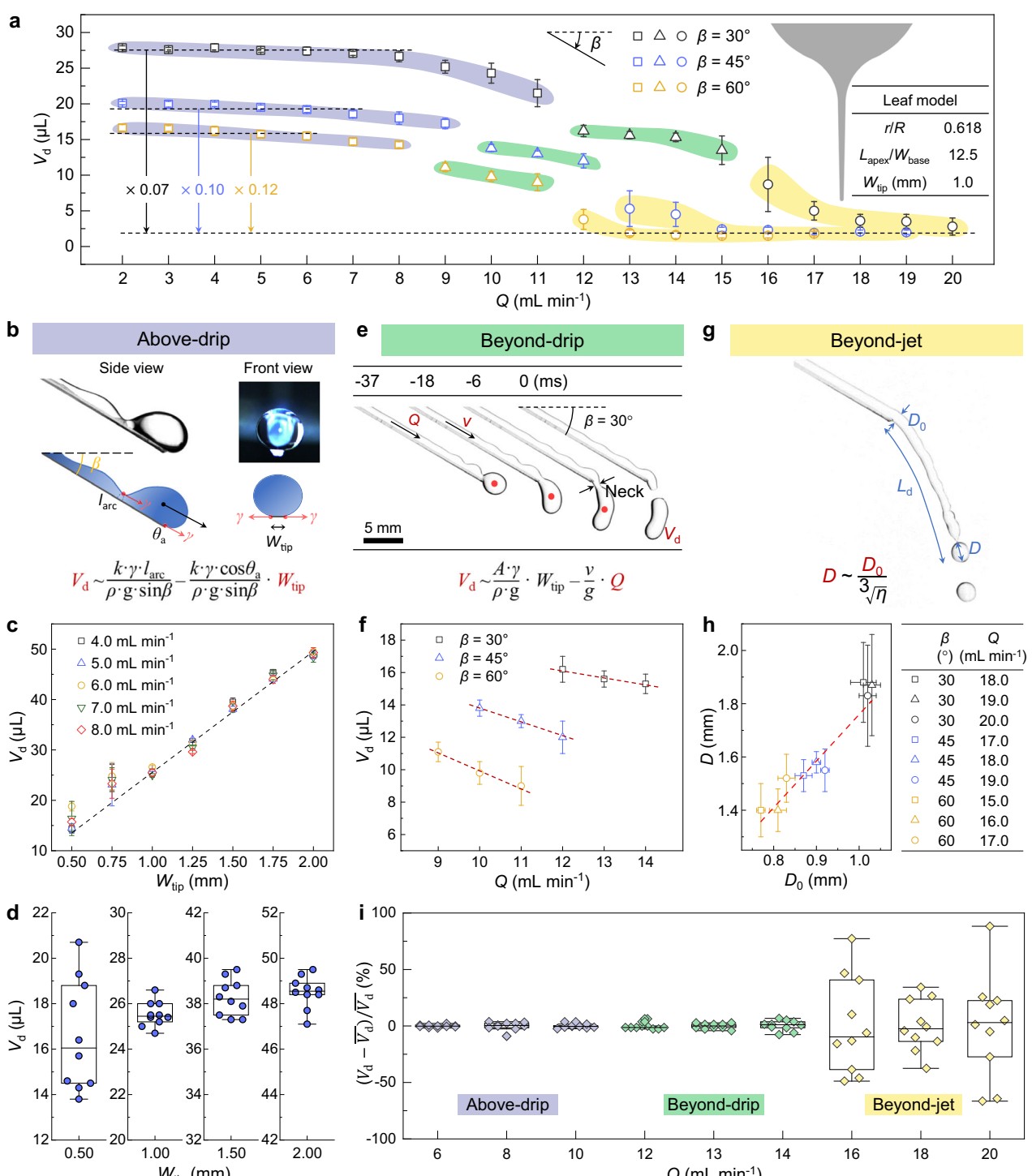

**Fig. 3 | Reduced drip volume and stabilized volume deviation of Beyond-drip state. a** Variation of drip volume $V_d$ at $Q$ of 2.0–20.0 mL min⁻¹ and $\beta$ of 30–60° of the optimized bodhi leaf apex with $r/R = 0.618$, $L_{apex}/W_{base} = 12.5$ and $W_{tip} = 1.0$ mm. **b** Experimental image and scheme of the Above-drip state. **c** $V_d$ scales with $W_{tip}$ (R² = 0.98) in the Above-drip state ($\beta = 30°$). Dashed line is linear fitting of $V_d$ values with $W_{tip}$ ranging from 1.0 mm to 2.0 mm. **d** Box chart of the $V_d$ deviation of different $W_{tip}$ values. **e** Drainage dynamics of the Beyond-drip state. $Q = 13.0$ mL min⁻¹ and $\beta = 30°$. **f** $V_d$ scales with $Q$ in the Beyond-drip state. **g** Image of the Beyond-jet state. The initial water filament diameter $D_0$ determines the drop diameter $D$. **h** $D$ scales with $D_0$ (R² = 0.85) in the Beyond-jet state. **i** The relative deviation of $V_d$, $(V_d - \bar{V_d})/\bar{V_d}$, of continuous ten droplets in one drainage process at a specific flow rate $Q$ ($\beta = 30°$). Beyond-drip state can achieve a smaller $V_d$ and minor deviation. Data in (**a**, **c**, **f** and **h**) are shown as mean ± SD, and the error bar represents SD ($n = 3$ independent experiments). For the box plots in (**d**, **i**), the bounds and center line of boxes show the 25/75 percentiles and median values, and the upper and lower whiskers show maxima and minima values ($n = 10$). Source data for (**a**, **c**, **d**, **f**, **h** and **i**) are provided as a Source Data file.

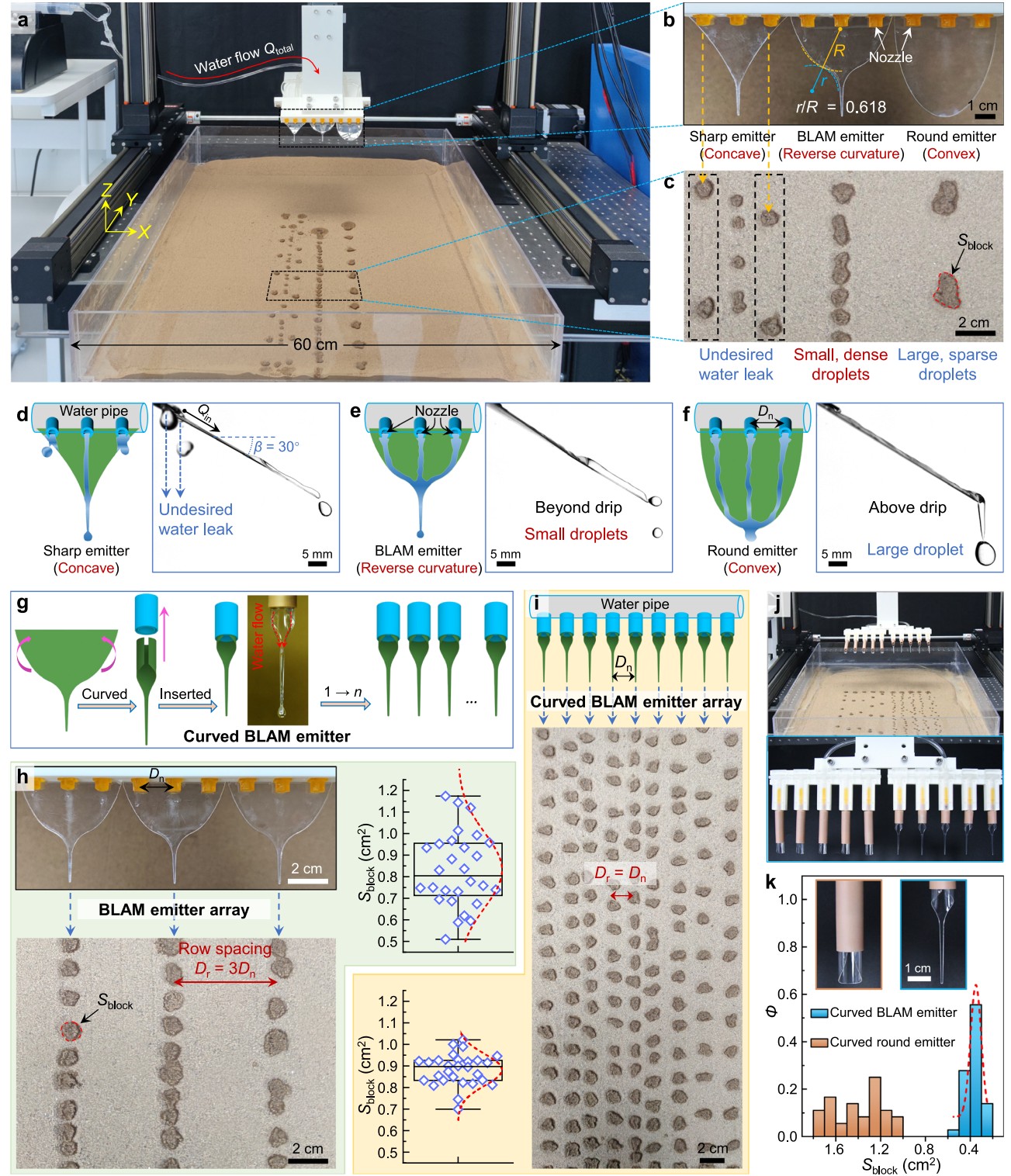

irrigation methods. Developing advanced planting and irrigation technology on barren land, such as sandy soil, can not only produce more food but also tackle challenges ranging from water resource stress to soil erosion[42]. The long-term evolution endows plants with functions of drainage, irrigation, and water transport[15,43,44]. Founded on the bodhi leaf enhanced dropwise drainage with high $f$ and low $V_d$, we can broaden its versatility in developing precise agricultural drip irrigation systems to solve current challenges.

Drip irrigation is the most widely used water-saving irrigation technology in arid areas and enables farmers to achieve higher yields

while saving water[45–47]. Water is delivered under pressure through a pipeline system to the crop fields and drips slowly onto the soil through emitters that are located close to the plants[48]. Based on the multi-axis displacement table, a mobile drip-irrigation prototype was designed (Fig. 4a). Biomimetic bodhi leaf apex was used as a water drop emitter and mounted onto the drip-irrigation pipeline, i.e., bodhi-leaf-apex-mimetic (BLAM) emitter (Fig. 4b, middle). Control experiments were also designed using a sharp emitter with concave curvature (Fig. 4b, left) and a round emitter with convex curvature (Fig. 4b, right). These two-dimensional (2D) emitters were mounted onto the

**Fig. 4 | Bodhi-leaf-apex-mimetic (BLAM) emitter-guided precise drip irrigation.**
**a** Drip-irrigation apparatus mounted on a multi-axis displacement table. **b** Optical image of drip irrigation pipeline with three different emitters, i.e., sharp emitter with concave curvature, BLAM emitter with reverse curvature $r/R = 0.618$, and round emitter with convex curvature. **c** Enlarged view of droplet pattern of three emitters in (**b**) on dry sandy soil after one drip irrigation process ($Q_{total} = 90.0$ mL min$^{-1}$) along the $Y$-axis. The moving speed of three emitters along the $Y$-axis, $v_Y$, is 75 mm s$^{-1}$. Sketch in top view and drainage dynamics in side view of three water flow on sharp emitter (**d**), BLAM emitter (**e**), and round emitter (**f**). $\beta = 30°$ and $Q_{in} = 30.0$ mL min$^{-1}$. **g** The manufacturing process of making a flat BLAM emitter into a curved BLAM emitter. **h** Optical image, drip irrigation pattern, and statistics of $S_{block}$ of flat BLAM emitter array. $S_{block}$ means the area of clumped sand

block. The irrigated row spacing $D_r$ equals $3D_n$, where $D_n$ is the distance between two neighboring nozzles. **i** Sketch, drip irrigation pattern, and statistics of $S_{block}$ of curved BLAM emitter array. The irrigated row spacing $D_r$ equals $D_n$. The moving speed of the emitters along the $Y$-axis, $v_Y$, is 75 mm s$^{-1}$ in (**h**) and (**i**). For the box plots in (**h**, **i**), the bounds and center line of boxes show the 25/75 percentiles and median values, and the upper and lower whiskers show maxima and minima values ($n = 28$). **j** The drip irrigation apparatus with curved round-emitter array (left) and curved BLAM-emitter array (right), which were used for the crop seedling growth experiments. **k** Probability density $\Phi$ of $S_{block}$ in the drip irrigation pattern image shown in (**j**) ($n = 36$ $S_{block}$ measured). The red dashed curves were the Gaussian fitting of $S_{block}$ histogram. Source data for (**h**, **i** and **k**) are provided as a Source Data file.

pipeline to converge water streams from three neighboring outlets into one merged water flow. The drip-irrigation pipeline moved along the $Y$-axis, driven by the multi-axis displacement table. The dripping water droplets from each emitter impacted the sandy soil and wet part of the soil zone (Fig. 4c and Supplementary Movie 4). After the impact process, water clumped the sand[49,50], forming pancake-like cratering blocks. The area of each sand block was termed $S_{block}$. Figure 4c compares the drip irrigation outcome of three emitters with different curvatures. Water streams could not be effectively aggregated on the sharp emitter and leaked from the concave emitter margin (Fig. 4d). Two-dimensional (2D) flat BLAM emitter with reverse curvature ($r/R = 0.618$) could realize desired smaller droplets and denser droplet patterns than the round emitter with convex curvature (Fig. 4c–f). The reverse curvature's effective convergence of water streams guarantees precise and controllable drip irrigation with high-frequency and small-volume characteristics.

According to FAO, water needs higher emitter discharge rates on sandy soils, and emitters thus have small waterways (typically 0.2–2.0 mm in diameter) to enhance water dripping frequency[48]. Such a small emitter diameter means that a significant hydrodynamic pressure needs to be overcome, and nozzle/emitter blockage will happen if the water contains sediments (Supplementary Fig. 14). Thus, developing a precise agricultural drip irrigation system with a larger nozzle to reduce flow resistance, blockage, and smaller dripping droplets at high frequency is eagerly expected yet very difficult with conventional agricultural irrigation systems.

Benefiting from the flexibility of PET substrate, the BLAM emitter can be used not only in a 2D flat manner but in a three-dimensional (3D) curved manner, where the 2D flat BLAM emitter can be curved and inserted into the nozzle on the irrigation pipeline, forming a curved BLAM emitter (Fig. 4g and Supplementary Fig. 15). Introducing a curved BLAM emitter could significantly reduce the drip volume compared to the original large nozzle without an emitter (Supplementary Fig. 16). Moreover, by switching the emitter shape between a 2D flat BLAM emitter and a 3D curved BLAM emitter, the irrigated row spacing can be easily regulated without modifying the original architecture of the irrigation pipeline (Supplementary Fig. 17 and Supplementary Movie 4). To be specific, the 2D flat BLAM emitter array can realize an aligned drip irrigation pattern with row spacing $D_r$ of $3D_n$, where $D_n$ is the distance between two neighboring nozzles (Fig. 4h). While in the case of the curved BLAM emitter array, a denser drip irrigation pattern with $D_r = D_n$ can be realized (Fig. 4i). The average blocked area $S_{block}$ was nearly the same in the flat BLAM emitter case and the curved BLAM emitter case, yet much less deviation in the latter case (Fig. 4h, i).

## Discussion

Proof-of-concept experiments on crop seedling growth were finally carried out to demonstrate the advantage of BLAM emitter for agricultural drip irrigation. A curved BLAM emitter array was chosen to guarantee controllable drip irrigation with high frequency and low drip volume characteristics. A curved round emitter array was also

fabricated for comparison (Fig. 4j, Supplementary Fig. 15). Compared with the curved round emitter, the curved BLAM emitter showed a higher water drip frequency and a denser irrigated coverage (Fig. 4j). For a curved round emitter, a significant drop ($V_d \approx 75$ μL, $R_0 \approx 2.6$ mm) ejected from the round emitter took about 9.0 ms to reach maximum spreading and finished retracting at 28.0 ms (Supplementary Fig. 18). In contrast, a small drop ($V_d \approx 15$ μL, $R_0 \approx 1.5$ mm) ejected from a curved BLAM emitter took about 4.0 ms to reach the maximum spreading area and finished retracting at 20.0 ms. The surface area of the final sand block, $S_{block}$, was more significant than the wetted area after retracting finished (Supplementary Fig. 18)[49]. The $S_{block}$ of a BLAM emitter, with a minor deviation, was about a quarter of that of a round emitter (Fig. 4k). The high drip frequency and small drip volume characteristics in curved BLAM emitter array promised the reduction of soil evaporation (Supplementary Fig. 19) and increased efficiency in drip irrigation[47,48].

For crop seedling growth, we first explored the effects of irrigation mode on the growth of wheat seedlings in sandy soil in laboratory environments (Supplementary Fig. 20, see "Methods"). Three conditions were compared, i.e., traditionally used border irrigation, curved round-emitter drip irrigation, and curved BLAM-emitter drip irrigation (Fig. 5a, Supplementary Fig. 20c, Supplementary Movie 5). First, water was poured directly onto the sandy soil in the border irrigation process. The sand particles severely clumped, forming a large block. The severely compacted sandy soil would delay wheat seeds from breaking out of the soil and cause slant growth of wheat seedlings (Fig. 5a, left). Then, in the round-emitter drip irrigation case, the sand compaction was slightly mitigated. However, the relatively large sand block still forced wheat slant growth (Fig. 5a, medium). As an advantage, in the curved BLAM-emitter drip irrigation process, the compacted sand block was tiny and hardly hindered wheat seedlings from sprouting and growing upright (Fig. 5a, right). The block, sprout, and slant ratios of the multiple replicates in the wheat seedling growth experiments were examined and quantified (Supplementary Fig. 21, Supplementary Movie 5, Supplementary Table 1). The curved BLAM-emitter drip irrigation exhibited the best drip irrigation behavior for wheat seedling growth, with the lowest block ratio, highest sprout ratio, and lowest slant ratio (Fig. 5b–d). The sandy soil under BLAM-emitter drip irrigation showed higher soil moisture than that of border irrigation and round-emitter drip irrigation, which accounted for a higher sprout ratio and growth rate in the case of BLAM-emitter drip irrigation (Supplementary Fig. 22).

Besides wheat, cotton was also grown on the sandy soil under two drip irrigation modes (Fig. 5e). The cotton seedlings exhibited a higher sprout ratio and growth height in a period of 21 days under the BLAM-emitter drip irrigation treatment (Supplementary Fig. 23, Supplementary Movie 6). Further, maize seedling growth results also showed higher sprout ratio and seedling height under BLAM-emitter drip irrigation than that under border irrigation[45] for both indoor (Fig. 5f, Supplementary Movie 7) and outdoor experiments (Supplementary Fig. 24, Supplementary Movie 8). Overall, the seedling growth experiments about wheat, cotton, and maize all confirmed that the

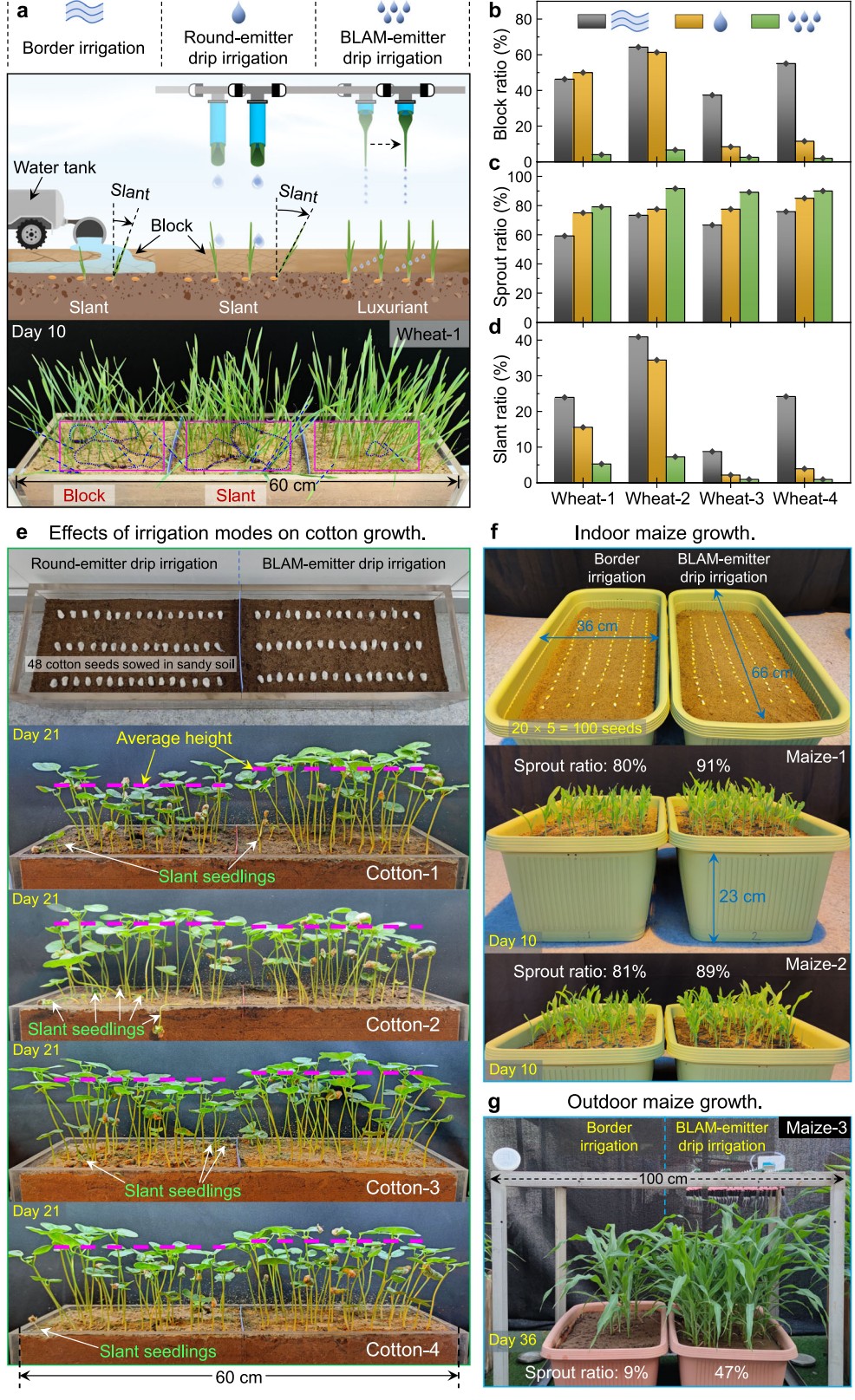

**a** Border irrigation | Round-emitter drip irrigation | BLAM-emitter drip irrigation

**e** Effects of irrigation modes on cotton growth.

**f** Indoor maize growth.

**g** Outdoor maize growth.

proposed BLAM-emitter drip irrigation could realize enhanced irrigation effect and boost crop growth in sandy soil, benefiting from the high $f$ and small $V_d$ characteristics. To further test the versatility of our BLAM-emitter drip irrigation, we constructed a small drip irrigation system, where the curved BLAM emitters were inserted into the soil close to the crop seedlings (Supplementary Fig. 25). The four kinds of crop seedlings grew well under BLAM-emitter drip irrigation for over

two months. Significantly, the BLAM emitter supplied water on demand and guaranteed the pitcher plant growth in the greenhouse for over one year (Supplementary Fig. 26).

In summary, the reverse curvature and long-tail enhanced drainage mechanisms of the bodhi leaf apex were revealed. The termed Beyond-drip state exhibited high drip frequency and small drip volume compared to the Above-drip state. We fabricated the BLAM-emitter

**Fig. 5 | BLAM-emitter drip irrigation boosts crop growth. a** Scheme of agricultural irrigation experimental setup with three irrigation modes, i.e., border irrigation (left), round-emitter drip irrigation (middle), and BLAM-emitter drip irrigation (right). For border irrigation, a certain amount of water was slowly poured onto sandy soil by a water pipe (inner diameter of 5 cm). For round-emitter drip irrigation and BLAM-emitter drip irrigation, the crop was irrigated by the mobile drip irrigation system equipped with a curved round emitter array and curved BLAM emitter array, respectively (see Fig. 4j and Supplementary Fig. 20c). Optical image in the bottom was the final growing condition of wheat seedlings under three irrigation modes after sowed in sandy soil for 10 days. The areas marked by purple rectangular boxes were chosen to evaluate the block ratio, where the block area was marked by blue dotted polygons. The blue dashed lines marked some of those slant seedlings. Variation of block ratio (**b**), sprout ratio (**c**) and slant ratio (**d**) of four groups of wheat growth experiments shown in Supplementary Fig. 21. The detailed statistics results can be found in Supplementary Table 1. **e** Indoor cotton seedling growth in sandy soil under two drip irrigation modes in a period of 21 days. The purple dashed lines indicated the average height of cotton seedlings. **f** Indoor maize seedling growth in sandy soil under border irrigation and BLAM-emitter drip irrigation. **g** Outdoor seedling growth experiments of maize seedlings over a period of 36 days. Source data for (**b**, **c**, and **d**) are provided as a Source Data file.

drip irrigation prototype based on the bodhi leaf apex shape, with optimized reverse curvature $r/R$ of 0.618, the golden section point. The proposed mobile BLAM-emitter drip irrigation prototype working at Beyond-drip state featured high drip frequency and small drip volume with minor relative deviation. It mitigated water shortage and soil compaction issues in agricultural applications, promising precise and sustainable agricultural irrigation. Future research should validate the use of BLAM emitters in a large-scale field setting. Although humans benefit greatly from biomimetic research[17,18], the natural plant, the product of 3.8 billion years of evolution with biodiversity, is under siege[51,52]. We, therefore, need to develop sustainable strategies in agricultural applications[53–55], such as the proposed BLAM-emitter drip irrigation strategy, to meet a sustainable future.

## Methods
### General information
Bodhi leaves used for statistical analysis were collected from three places: part of them were directly taken from young bodhi trees self-grown in Beijing (40°N, 116°E), China; others were purchased from the garden in Jieyang, Guangdong (23°N, 116°E) and Yulin, Guangxi (23°N, 110°E), China. The PET (thickness 300 μm) and Cu (thickness 200 μm) substrates were purchased from Alibaba.com online. The PET-based artificial bodhi leaf samples were prepared by laser cutting (LSC30 $CO_2$ laser, HGTECH, China). Cu-based bodhi leaf apex samples were prepared by laser cutting (LSF20 laser, HGTECH, China). The PET- and Cu-based samples were flushed with ethanol and water and dried with $N_2$ gas before use. Field-emission SEM SU-8010 (Hitachi, Japan) captures the scanning electron microscopy of PET substrates with an accelerating voltage of 10 kV. The light microscopy (Olympus DSX1000, Japan) captured the optical morphology of Cu substrates. Contact angles were measured with 2.0 μL water droplets using an OCA20 contact angle measurement equipment (DataPhysics, Germany). A high-speed camera (PHOTRON FASTCAM Mini UX100, Japan) recorded the water drainage dynamics at 1000–5000 frames per second (fps). A high-speed camera (PHOTRON FASTCAM Mini WX100, Japan) recorded the dynamics of water droplets impacting sandy soil at 4000 fps. A micro-injection gear pump (Harvard Pump, USA) controlled the water flow rate $Q$. All the water used in this work was deionized water (18.2 MΩ cm) from Milli-Q equipment. The water flow speed $v$ was obtained using the Photron Fastcam Viewer software and processed using Excel 365 (Microsoft) and Origin 2021 (OriginLab). The mobile drip-irrigation prototype was built on a homemade multi-axis displacement table equipped with linear stages (RXP45-200, QRXQ, China). The soil evaporation experiments were performed using a solar simulator (SS-F7-3A, Enlitech). Photographs of bodhi leaves, crops, and drip-irrigation experimental setup were recorded by a digital camera (Nikon D750, Japan). The soil moisture was measured by a wireless soil parameter sensor (JXCT, China).

### Water drainage experiments on artificial leaf model
PET-based or Cu-based artificial leaf model was mounted at a fixed inclination angle $\beta$ (the angle between the leaf surface and the horizontal). High-speed cameras captured the water drainage dynamics from top and side views. A micro-injection gear pump controlled the water flow rate. Figure 2b shows the three-needle experimental setup, where the three-orifice needle (outer diameter 0.8 mm, inner diameter 0.5 mm) was mounted just above and parallel to the leaf model. The distance between the central needle tip and the artificial leaf apex tip was fixed at 30.0 mm. Figure 2f shows the single-needle experimental setup, where the needle was always vertically mounted regardless of $\beta$. For water drainage experiments in Fig. 2f–h and Fig. 3, the distance between the needle tip and the artificial leaf apex tip was fixed at "$L_{apex} + 15.0$" mm.

### Calculation of drip frequency $f$ and drip volume $V_d$
Typically, the high-speed camera recorded water drainage dynamics with a lasting time $t_0$ of 4–6 s. Drip frequency $f$ was calculated as $f_N = N_{drop}/t_0$ by counting the number of dripping droplets, $N_{drop}$. For single-needle dripping experiments in the Above-drip and Beyond-drip states (Fig. 3b–f), the drainage process exhibited a precise period $T_{drip}$ (the time between two continuous dripping droplets). In that case, the drip frequency $f$ could be calculated as $f_T = (T_{drip})^{-1}$, and here, the value of $f_T$ was approximately equal to $f_N$. The drip volume $V_d$ was measured geometrically by fitting a specific droplet shape with a cylindrical-symmetry circle or ellipse. Mainly, if the dripping period $T_{drip}$ was exact (with the relative deviation of $T_{drip}$ less than 5%) in one drainage process, $V_d$ could be deduced as $Q \cdot T_{drip}$ or $Q \cdot (f_T)^{-1}$.

### Measurement of water flow speed $v$ on artificial bodhi leaf apex
We performed particle image velocimetry experiments to evaluate the water flow speed $v$ on the leaf apex. Nylon particles (average diameter 50 μm) were chosen as tested particles to reflect the water flow speed on the leaf apex, which could suspend in water and follow the water flow. The drainage dynamics of nylon-particle-contained water on the bodhi leaf apex were recorded from both top and side views at 2000 fps (Supplementary Fig. 6). It should be noted that the addition of nylon particles hardly affected the drainage frequency $f$ and drip volume $V_d$ in the drainage experiments. The water flow speed $v$ was tracked using the Photron Fastcam Viewer software. At least 20 nylon particles were chosen and measured for each test to give the average water flow speed. We deliberately selected those nylon particles suspended in the middle of the water flow and flew forward parallel with the apex (Supplementary Movie 1) to reduce the measurement error.

### Critical water flow speed for breaking contact line pinning
We measured the critical flow speed $v_c$ for the Beyond-drip state in both three-needle and single-needle systems (Supplementary Fig. 9). Once the flow speed exceeds $v_c$, the water flow could break the contact line pinning at the apex tip and realize the Beyond-drip state. The critical speed $v_c$ measured in the three-needle setup and the single-needle setup (Supplementary Fig. 9) was constant and in the same value range of 230–260 mm s$^{-1}$, corresponding to flow rates $Q$ range of 16.0–36.0 mL min$^{-1}$ in the three-needle experimental setup (Supplementary Fig. 9a, b) and 8.0–16.0 mL min$^{-1}$ in the single-needle experimental design (Supplementary Fig. 9d, e).

### Wettability modification of PET-based artificial bodhi leaf apices

The untreated PET sample showed an intrinsic water contact angle (WCA) of 76.1° ± 1.9°. For hydrophilic PET-based bodhi leaf apex, the sample was treated with $O_2$ plasma (DT-03, OPS Plasma Technology, China) at 200 W for 2 min, yielding WCA of 34.1° ± 3.3°. For hydrophobic PET-based bodhi leaf apex, the sample was treated with fluorosilane in a vacuum at 80 °C for 12 h, yielding WCA of 114.2° ± 4.2°. For superhydrophobic PET-based bodhi leaf apex, the sample was immersed in superhydrophobic coating and taken out alternatively 3 times, yielding WCA of 150.2° ± 1.7°.

### Crop seedling growth experiments

The experiments on crop seedling growth were performed in Beijing (40°N, 116°E), China. The sandy soil and crop seeds were purchased from Handan (36°N, 114°E), Hebei Province, China, without pre-treatments. The seedling growth of wheat and cotton was performed in the laboratory, and the maize seedling growth was performed in the laboratory and outdoors. During the indoor seedling growth, the simulated light source was provided to the crop seedlings for 12 h daily (from 9:00 to 21:00). The temperature and relative humidity (RH) were recorded daily by a hygrothermograph during the crop seedling growth (June and July 2023).

For wheat growth, the wheat seeds were randomly divided into three groups, each including 120 seeds, and sowed into four rows (30 seeds per row) about 3.0 cm beneath the sandy soil surface (see Supplementary Fig. 21). Wheat seedlings were planted in an acrylic-based planting box (inner size 60 × 15 × 10 cm, length × width × depth). The wheat seedlings were watered every two days. About 100.0 mL water was poured each time onto the sandy soil surface using a large pipe (inner diameter 5 cm) for border irrigation. For round-emitter drip irrigation and BLAM-emitter drip irrigation, the wheat seeds were irrigated using the mobile drip irrigation system shown in Fig. 4j. The total irrigated water quantity was about 100.0 mL each time for round-emitter drip irrigation and BLAM-emitter drip irrigation. The wheat seedling growth was recorded from the front view using a time-lapse camera (aTLi Eon T100, China) at a 10-min time interval. The growing condition of wheat seedlings under different irrigation modes was analyzed on the 10th day after sowing.

For cotton growth, the cotton seeds were randomly divided into two groups, each including 48 seeds, and sowed into three rows (16 seeds per row) about 3.0 cm beneath the sandy soil surface. The cotton seedlings were grown in the same acrylic-based planting box as wheat seedlings. The cotton seedlings were irrigated with 150.0 mL water every two days under two drip irrigation modes using the mobile drip irrigation apparatus, consistent with those in wheat growth. For indoor and outdoor maize growth, we randomly selected 100 maize seeds and sowed them into the sandy soil in a rectangular pot (Fig. 5f, g, Supplementary Fig. 24, Supplementary Movies 7 and 8). The seeds were planted into five rows (20 seeds per row) and buried about 5.0 cm beneath the soil surface. Typically, irrigation was supplied approximately every 4 days for maize seedling growth. The quantity of irrigated water was about 1.2 L for both border irrigation and BLAM-emitter drip irrigation each time.

### Reporting summary

Further information on research design is available in the Nature Portfolio Reporting Summary linked to this article.

## Data availability

All data needed to evaluate the conclusions in the paper are available in the main text or the supplementary materials. Source data are provided with this paper.

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

## Acknowledgements

We acknowledge project funding provided by the National Natural Science Foundation (22122508 to Z.D., 52173293 to Z.D.), the National Key Research and Development Program of China (2021YFA0716702 to C.Y.), and the Young Elite Scientists Sponsorship Program by China Association for Science and Technology (Z.D.).

## Author contributions

Z.D. conceived the idea and designed the research. S.L., T.S., Z.Z., J.P., and C.Y. performed the experiments. S.L., C.Z., L.J., and Z.D. analyzed the data. S.L. and Z.D. wrote the manuscript. All the authors discussed the results and commented on the manuscript.

## Competing interests

The authors declare the following competing interests. Two patent applications have been filed by the Beijing Institute of Bionic Interface Science and Future Technology based on the drip irrigation device described here (China Patent Application Serial No. ZL 202223393147.0, ZL 202223393143.2), on which S.L., C.Y., Z.D., and L.J. are listed as inventors. The remaining authors declare no competing interests.
