## [Peer Review File · Nature Communications]

Reviewers' Comments:

Reviewer #1:

Remarks to the Author:

In this manuscript, the authors developed a fancy drip irrigation system inspired by the Bodhi leaf apex. In particular, the effects of the reverse curvature and the long tail of leaf apices on drainage dynamics were thoroughly demonstrated in a logical manner. First, screening pre-experiments were performed to understand the drainage mechanism and quantified the threshold parameters of L/D and r/R for the drainage experiments and drip-irrigation demonstration. Second, the plant experiments verified the beneficial effects of the designed BLAM-type drip irrigation system. The understanding of the natural idea and the optimization design and screening of the biomimetic materials innovate the application of drip-irrigation systems with spatial and temporal scales. Overall, this manuscript offers new ideas, advances, and significant contributions to the field of biomimetic research and agricultural irrigation, and this work will be of great interest to the related community for further studies of drainage and drip irrigation systems. Therefore, I recommend the manuscript for publication in Nature Communications. Below are some minor recommendations to help improve the manuscript.

1. Line 97. The authors may want to add the microscope morphology and wettability of natural Bodhi leaf. Do the natural and artificial Bodhi leaves share similar wettability?
2. Does wettability have any influence on drainage behavior?
3. Line 200. The authors focused on the breakup of water flow at the leaf tip. The results of the three-needle and single-needle experiments have a uniform breakup speed. The restriction of the water-solid contact line at the edge should effect. I wonder what the microstructure of the artificial leaf tip is. Is it flat, or does it have other microstructure(s)? The authors should give morphology details about the leaf tip and margin in Fig. 2A.
4. Line 325 What's the moving speed (range) of your drip emitters along Y-axis in Figure 4?

Reviewer #2:

Remarks to the Author:

1. I read the paper titled 'Reverse curvature enhanced drainage of bodhi leaf for agricultural drip irrigation'. The paper revealed the drainage mechanism of the Bodhi leaf and described the reverse curvature of the leaf shape the and long-tail leaf apex which contributes to fast water drainage. It is one of the noteworthy results of this paper.
2. The relationship between drip frequency and leaf shape and apex features was revealed. It is another noteworthy result of this paper.
3. The authors applied the optimal bodhi leaf apex to drip irrigation, designed a drip emitter, and conducted irrigation experiments with the drip emitter. It's a new drip emitter, unlike existing drip emitters (online and inline drippers), while the practical is a question worth discussing. In agricultural drip irrigation, water is transported to fields under pressure. For the BLAM drip emitter, how to dissipate energy?
4. I cannot entirely agree with the results of wheat and cotton seedlings. For the three irrigation modes, i.e., border irrigation, round drip irrigation, and the BLAM drip irrigation, the irrigation water amount is not same, but the emergence and growth of seedlings are depended on soil moisture.

Response to Reviewers

Manuscript:

Reverse curvature enhanced drainage of Bodhi leaf for agricultural drip irrigation

This PDF file includes:

Responses to Reviewer # 1.....	2
Responses to Reviewer # 2.....	6

Reviewer comments are set on grey background, and responses to the comments are shown in blue and newly added text in the revised manuscript and SI are highlighted by yellow.

Responses to Reviewer # 1

Reviewer's comments:

In this manuscript, the authors developed a fancy drip irrigation system inspired by the Bodhi leaf apex. In particular, the effects of the reverse curvature and the long tail of leaf apices on drainage dynamics were thoroughly demonstrated in a logical manner. First, screening pre-experiments were performed to understand the drainage mechanism and quantified the threshold parameters of L/D and r/R for the drainage experiments and drip-irrigation demonstration. Second, the plant experiments verified the beneficial effects of the designed BLAM-type drip irrigation system. The understanding of the natural idea and the optimization design and screening of the biomimetic materials innovate the application of drip-irrigation systems with spatial and temporal scales. Overall, this manuscript offers new ideas, advances, and significant contributions to the field of biomimetic research and agricultural irrigation, and this work will be of great interest to the related community for further studies of drainage and drip irrigation systems. Therefore, I recommend the manuscript for publication in Nature Communications. Below are some minor recommendations to help improve the manuscript.

Response: Many thanks for the reviewer's positive evaluation of our manuscript and constructive comments. We have added new morphological characterization, drainage experiments, and corresponding discussions according to the valuable suggestions. We have addressed the comments point-by-point as below.

1. Line 97. The authors may want to add the microscope morphology and wettability of natural Bodhi leaf. Do the natural and artificial Bodhi leaves share similar wettability?

Response: Thanks for the reviewer's valuable suggestions. We have added the morphological and wettability characterization of natural Bodhi leaf in **Supplementary Fig. 2** (page 3, lines 16-20) in the revised supplementary information. The natural Bodhi leaf showed water contact angle (CA) of $86.6^\circ \pm 2.4^\circ$, a little higher than those of PET-based ($76.1^\circ \pm 1.9^\circ$) and Cu-based ($68.7^\circ \pm 1.7^\circ$) artificial Bodhi leaves. Nevertheless, the water CA values of natural and artificial Bodhi leaves were all located in the CA range of $30^\circ \sim 110^\circ$, a suitable water contact angle range for continuous and stable drainage process on Bodhi leaf apex (**Supplementary Fig. 10** in the revised supplementary information, see below).

Newly added **Supplementary Fig. 2.** Wettability and morphology characterization of natural Bodhi leaf. **a** Optical image of a natural Bodhi leaf. **b** Water contact angle on the adaxial surface of natural Bodhi leaf. **c** Light microscopy of natural Bodhi leaf apex. The scanning electron microscopy of natural Bodhi leaf at adaxial surface (**d**), leaf margin (**e**), and apex tip (**f**).

2. Does wettability have any influence on drainage behavior?

Response: Thanks for the reviewer's comments. To figure out the effect of leaf apex wettability on drainage behavior, we performed water drainage experiments using the optimized PET-based Bodhi leaf apex ($r/R = 0.618$, $L_{\text{apex}}/W_{\text{base}} = 12.5$) with different water contact angles (CAs). Apart from untreated PET substrate (with CA of $76.1^\circ \pm 1.9^\circ$), three other PET-based leaf apex samples were fabricated, *i.e.*, hydrophilic PET-based leaf apex (with CA of $34.1^\circ \pm 3.3^\circ$), hydrophobic PET-based leaf apex (with CA of $114.2^\circ \pm 4.2^\circ$), and superhydrophobic PET-based leaf apex (with CA of $150.2^\circ \pm 1.7^\circ$). Taking $\beta = 45^\circ$ as an example (**Supplementary Fig. 10** in the revised manuscript). In the case of hydrophilic, untreated, and hydrophobic PET substrates, the injected water formed a continuous liquid stream on PET-based leaf apices and exhibited Above-drip state at Q of $5.0 \sim 8.0 \text{ mL} \cdot \text{min}^{-1}$. The drip volume V_d was stable and generally close. When it came to superhydrophobic PET-based leaf apex, the injected water flow broke up and formed droplets near the water flow injected point, leading to a discontinuous and unstable drainage process. Therefore, our artificial Bodhi leaf apex performed drainage behaviors well under a suitable CA range of $30^\circ \sim 110^\circ$.

Newly added **Supplementary Fig. 10**. Effects of wettability on drainage behavior of PET-based Bodhi leaf apex. Water contact angle and corresponding drainage dynamics on hydrophilic (a, b), untreated (c, d), hydrophobic (e, f), and superhydrophobic (g, h) PET-based leaf apices. i Variation of drip volume V_d with injection flow rate Q at β of 45° under three wettability states.

3. Line 200. The authors focused on the breakup of water flow at the leaf tip. The results of the three-needle and single-needle experiments have a uniform breakup speed. The restriction of the water-solid contact line at the edge should effect. I wonder what the microstructure of the artificial leaf tip is. Is it flat, or does it have other microstructure(s)? The authors should give morphology details about the leaf tip and margin in Fig. 2A.

Response: Thanks for the reviewer’s suggestions. The microscope morphology of the PET-based artificial leaf apex has been added in the updated Fig. 2a in the revised manuscript. The central part of the PET-based artificial Bodhi leaf was smooth. The margin was bulged forming a ridge with a width of $\sim 80 \mu\text{m}$, due to the fusing effect in the laser-cutting process. Such a ridge functioned to resist water flow and pin the three-phase contact line at PET-based leaf apex tip, leading to the Above-drip state at low water flow rate.

Updated **Fig. 2a**. Optical image of PET-based biomimetic Bodhi leaf with shape parameters R , r , W_{base} , L_{apex} , and W_{tip} noted from top to bottom. Reverse curvature, r/R , controls convergence region and long tail, $L_{\text{apex}}/W_{\text{base}}$, controls drainage region. Insets are the contact angle of a water droplet on

the PET surface and scanning electron microscopy (SEM) images of the PET substrate in the central part and at apex tip.

4. Line 325. What's the moving speed (range) of your drip emitters along Y-axis in Figure 4?

Response: Thanks for the reviewer's comments. The moving speed along Y-axis, v_Y , ranges from 0 to 100 mm/s on demand. In Fig. 4**a-c**, **h**, and **i**, v_Y was about 75 mm/s. We have added this value in the legend of Fig. 4 on pages 18-19, line 386, 393 in the revised manuscript.

Responses to Reviewer # 2

Reviewer's comments:

1. I read the paper titled 'Reverse curvature enhanced drainage of Bodhi leaf for agricultural drip irrigation'. The paper revealed the drainage mechanism of the Bodhi leaf and described the reverse curvature of the leaf shape and long-tail leaf apex which contributes to fast water drainage. It is one of the noteworthy results of this paper.

Response: We appreciate the reviewer for the valuable assessment of our manuscript.

2. The relationship between drip frequency and leaf shape and apex features was revealed. It is another noteworthy result of this paper.

Response: Many thanks for the reviewer's positive evaluation of our manuscript.

3. The authors applied the optimal bodhi leaf apex to drip irrigation, designed a drip emitter, and conducted irrigation experiments with the drip emitter. It's a new drip emitter, unlike existing drip emitters (online and inline drippers), while the practical is a question worth discussing. In agricultural drip irrigation, water is transported to fields under pressure. For the BLAM emitter, how to dissipate energy?

Response: Thanks for the reviewer's comments. The proposed BLAM emitter was indeed different from existing drip emitters. Just as the reviewer states, water flow is transported along the pipeline under pressure. In our experiments, the inner diameter of the curved BLAM emitter was larger, which means flow resistance is lower than that of existing drip emitters. In order to keep the flow rate the same at each emitter along the whole pipeline, we introduced a commercial valve to balance the pipe pressure (**Supplementary Fig. 24** in the revised manuscript). As an advantage, the reverse curvature of the leaf shape and long-tail leaf apex contributes to fast water drainage. Therefore, our proposed BLAM emitter could realize high-frequency and small-volume drip irrigation processes under a stable flow rate, boosting the emerging and upright growth of crop seedlings.

Newly added **Supplementary Fig. 24**. Outdoor maize cultivation experiments. **a** Growth condition of maize seedlings under border irrigation and BLAM-emitter drip irrigation after sowed in sandy soil 36 days. **b** The curved BLAM-emitter array for drip irrigation. **c** The structure of one curved BLAM-emitter.

4. I cannot entirely agree with the results of wheat and cotton seedlings. For the three irrigation modes, i.e., border irrigation, round drip irrigation, and the BLAM drip irrigation, the irrigation water amount is not same, but the emergence and growth of seedlings are depended on soil moisture.

Response: We greatly appreciate the reviewer for the valuable comments. We have performed multiple replicates in the cultivation experiments of wheat, cotton, and maize (updated **Fig. 5b-g**, newly added **Supplementary Fig. 20 - 24**). In these new experiments, the irrigation water amount was kept the same among different irrigation modes for specific crop cultivation. And the new experiment results consolidated the previous conclusion that the BLAM-emitter drip irrigation showed best performance towards crop cultivation than border irrigation and round-emitter drip irrigation. Besides, we measured soil moisture at day 3, 5, and 7 during the wheat growth process (**Supplementary Fig. 22** in the revised manuscript). The sandy soil under BLAM-emitter drip irrigation showed higher soil moisture than that of border irrigation and round-emitter drip irrigation. High soil moisture benefits the emergence and growth of seedlings, which accounts for the best crop cultivation performance by BLAM-emitter drip irrigation.

Newly added **Supplementary Fig. 22**. Measurement of soil moisture during the wheat growth process. **a** The used wireless soil parameter sensor, which has three probes (6 cm length) and can measure soil moisture immediately after inserting the probes into the soil. Soil moisture measurement process on soil surface under border irrigation (**b**), round-emitter drip irrigation (**c**), and BLAM-emitter drip irrigation (**d**). **e** Statistics results of soil moisture measurement during the wheat growth process. Two points were measured for each irrigation zone in one measurement process. The measurement was performed every two days. The sandy soil under BLAM-emitter drip irrigation showed higher soil moisture than that of border irrigation and round-emitter drip irrigation.

Reviewers' Comments:

Reviewer #1:

Remarks to the Author:

The authors have thoroughly addressed and rectified the issues present in the second manuscript. Therefore, I am in agreement with the publication of this manuscript in this journal.

Reviewer #2:

Remarks to the Author:

The authors addressed my concerns and improved the MS.

In the section of Discussion, the authors should discuss the future application of the BLAM emitter and its shortcomings.

I suggested that it could be accepted after minor revision.

Response to Reviewers

Manuscript:

Efficient agricultural drip irrigation inspired by fig leaf morphology

This PDF file includes:

Responses to Reviewer # 1.....2

Responses to Reviewer # 2.....3

Reviewer comments are set on grey background, and responses to the comments and newly added text in the revised manuscript are shown in blue.

Responses to Reviewer # 1

Reviewer's comments:

The authors have thoroughly addressed and rectified the issues present in the second manuscript. Therefore, I am in agreement with the publication of this manuscript in this journal.

Response: Many thanks for the reviewer's support and insightful suggestions to improve the quality of our manuscript.

Responses to Reviewer # 2

Reviewer's comments:

The authors addressed my concerns and improved the MS.

Response: Thanks for the reviewer's positive evaluation.

In the section of Discussion, the authors should discuss the future application of the BLAM emitter and its shortcomings.

Response: Thanks for your comment. We have added the prospect of the BLAM emitter on page 15, lines 363-365 in the revised manuscript.

Our modification to the manuscript: On page 15, lines 363-365, we added a sentence discussing the future application of the BLAM emitter, "Further attempts are needed to expand the application scope of the BLAM emitters, for example, BLAM emitter for intelligent response to soil moisture, and pressure compensated BLAM drip emitter."

I suggested that it could be accepted after minor revision.

Response: Many thanks for your support and assistance through the peer review process.